

# Drivers and assemblies of soil eukaryotic microbes among different soil habitat types in a semi-arid mountain in China

He Zhao*, Xuanzhen Li*, Zhiming Zhang, Yong Zhao, Peng Chen and Yiwei Zhu

College of Forestry, Henan Agricultural University, Zhengzhou, China
* These authors contributed equally to this work.

## ABSTRACT

The effects of environmental and species structure on soil eukaryotic microbes inhabiting semi-arid mountains remain unclear. Furthermore, whether community assembly differs in a variety of soil habitat types, for example, artificial forest, artificial bush, farmland, and natural grassland, is not well understood. Here, we explored species diversity and composition of soil eukaryotic microbes south of the Taihang Mountains (mid-western region of China) using Illumina sequencing of the 18S rRNA gene (V4) region on the MiSeq platform. The results suggest that the forest soil habitat type improved the diversity and abundance of soil eukaryotic microbes that will benefit the restoration of degraded soil. The SAR (Stramenopiles, Alveolates, Rhizaria) supergroup and Metazoa were the dominant soil eukaryotic microbial groups at the phylum level. About 26% of all operational taxonomic units were common among the different soil habitat types. The O-elements, water content, soil organic matter, and elevation significantly influenced the abundance of soil eukaryote communities ($P < 0.05$). Our findings provide some reference for the effectiveness of local ecological restoration and the establishment of a soil eukaryotic microbe resource databases in a semi-arid area.

## INTRODUCTION

Eukaryotic microorganisms comprise most of the natural microbes and they are closely linked with the sustainability of the soil-based ecosystem and biogeochemical processes (*Coleman, Crossley & Hendrix, 2004*; *Falkowski, Fenchel & Delong, 2008*; *Delong, 2009*). The soil eukaryotic microbial group plays a key role in litter decomposition, nutrient cycling, and soil structural formation (*Brussaard, Ruiter & Brown, 2007*). However, the majority of microorganisms are still unknown and cannot be isolated from complex environmental matrices or are not cultivable on microbiological media (*Amaral Zettler et al., 2002*; *Rappé & Giovannoni, 2003*; *Bonkowski, 2004*), so they require detailed studies. Particularly, knowledge of the role and dynamics of soil eukaryotic communities in semi-arid areas is rather fragmentary.

Corresponding author
Yong Zhao,
zhaoyonghnnd@163.com

Species richness and diffusion capacity are features of eukaryotic microbial communities, which lead them to inhabit a variety of habitat types (*Cutler et al., 2013*; *Luria, Ducklow & Amaral-zettler, 2014*; *Niederberger et al., 2015*). Scholars have argued that the abundance of nematode, amoeba, and fungal communities generally increase in aquifer environmental habitats (*Madsen, Sinclair & Ghiorse, 1991*; *Novarino et al., 1997*), and several studies have demonstrated that organic fertilization strongly affects eukaryotic community composition in agricultural habitats (*Dong et al., 2014*; *Lentendu et al., 2014*; *Murase et al., 2015*). *Cutler et al. (2013)* showed that substrate characteristics affect the abundance/biomass of eukaryotic microbial communities but do not influence diversity in sandstone habitats. Most previous studies only focused on changes in communities of a single habitat type (*Allen et al., 2009*; *Sonjak, 2005*). However, the diversity and composition of eukaryotic microbial communities may differ among various soil habitat types, and comparative analyses of eukaryotic microbial communities are still rare in different semi-arid soil habitats.

Elucidating the composition and diversity of eukaryotic microbial communities is a crucial step for conserving soil productivity and sustainability (*Bass & Cavaliersmith, 2004*; *Staay et al., 2010*; *Czechowski et al., 2016*). Thus, it is necessary to understand the processes responsible for the distribution of eukaryotic microorganisms. Most attempts to identify the structure and coexistence mechanism of natural communities are the joint result of niche theory, which focuses on deterministic processes based on habitat patterns but is restricted by various factors (*Tokeshi, 1990*; *Jongman, Braak & Tongeren, 1995*; *Sarah et al., 2016*). Niche theory emphasizes that species composition is correlated closely with changes in environmental factors (*Leibold, 1995*). Some studies have supported niche differentiation and confirmed that abiotic soil factors structure microbial communities (*Dumbrell et al., 2010*; *Lennon et al., 2012*). Thus, environmental heterogeneity and spatially structured differences are key factors to explore the distribution of eukaryotic microorganisms. Although ecologists have conducted a series of explorations (*Elloumi et al., 2009*; *Volant et al., 2016*), the major environmental drivers of soil eukaryotic microorganism distribution in semi-arid regions remain unclear.

Obtaining detailed and accurate data is key to studies of soil eukaryotic microorganisms. However, traditional soil biodiversity assessments are fundamentally flawed, and the time consumed and taxonomic expertise severely limit the sorting and identification of eukaryotic microbiota (*Oehl et al., 2004*; *Velasco-Castrillón & Stevens, 2014*). Thus, new measurement and evaluation methods are constantly being updated. High-throughput sequencing (pyrophosphate sequencing with the Illumina Genome AnalyzerIIx) of soil samples affords an interesting opportunity to confirm microbial genetic variation and classification (*Margulies et al., 2006*; *Chown et al., 2015*; *Zhao et al., 2017*). This technology provides more comprehensive and accurate sequencing data (*Opik et al., 2010a*) that are beneficial to accurately identify microbes in a variety of vegetation types (*Rogers, 2007*; *Marchesi & Ravel, 2015*). Applying new technologies provides very useful insight, but more explicit microbiological information in a variety of research areas is still needed.

The area south of the Taihang Mountains in China has a typical semi-arid climate, and soil degradation is common (*Zhao, 2007*). Vegetation restoration is a traditional practice widely used to prevent ecosystem degradation in this region of China, and many areas have been planted with artificial shrubs and trees (*Zhao et al., 2017*). Hitherto, most studies of local vegetation restoration have focused on soil physicochemical properties, enzyme activities, root characteristics, and plant growing conditions (*Zhang, Xi & Li, 2006*; *Pei et al., 2018*; *Song et al., 2017*), but few studies have considered the characteristics of local microorganisms. We hypothesized that the presence of soil eukaryotic microbes is closely related to soil habitat types. To test our hypothesis, we used 18S rRNA gene (V4) region to analyze the sequenced data from the Illumina MiSeq platform. The test aimed to (1) identify the composition and richness of the current soil eukaryotic microorganisms in the area south of the Taihang Mountains; (2) illustrate any differences in soil eukaryotic microbes among soil habitat types, including forest, shrub, grass, and farmland soil habitats; (3) and assess the drivers of environmental factors on soil eukaryotic microorganism assembly in this semi-arid region.

## MATERIALS AND METHODS

### Study area

The area south of the Taihang Mountains (112°28′–112°30′E, 35°01′–35°03′N) is a semi-arid region in China. The study area has a continental monsoon climate in most parts, with an average annual sunshine rate of 54%, average air temperature of 14.3 °C, and annual precipitation of 440–860 mm. The most common soil is Ustalf (United States Department of Agriculture (USDA) classification). Local vegetation restoration began in 1958. The main land-use methods are grass soil, forest soil, shrub soil, and farmland. Most of the forest soil habitat types are artificial forests, including *Platycladus orientalis* (L.) Franco, *Quercus variabilis* Bl., and *Robinia pseudoacacia* L. *Lespedeza bicolor* Turcz., *Vitex negundo* L., and *Ziziphus jujuba* Mill. var. as the major shrub types. *Artemisia princeps* H. Lév., *Setaria viridis* (L.) Beauv., and *Arthraxon hispidus* (Thunb.) Makino are the dominant herbaceous plants in the natural grass soil. All farmland species were planted by local residents and include *Zea mays* L., *Triticum aestivum* L., *Ipomoea batatas* L., *Brassica campestris* L., and *Lycopersicon esculentum* Mill.

### Sampling design

Based on the status of the local vegetation resources (*Zhao, 2007*), we established study plots, which represented the ecological systems classified as the forest, shrub, grass, and farmland soil habitat types in October 2017. Samples were collected from the different soil habitat types (each type had three plots), and 12 independent plots (10 × 10 m) were set up. Soil samples were collected at a soil depth of 5–10 cm and were taken along the "S" model from five sampling points, then pooled together as one large sample for each plot. After cleaning up the impurities, 200 g soil in the O horizon was collected at a depth of zero to five cm. Root samples in the vicinity of the soil samples were also collected with a soil auger (inner diameter = 4 cm) at a depth of 5–10 cm. We carefully

collected the soil samples and placed them in freezing boxes for storage at $-70\,°C$ until further soil geochemical and molecular biological analyses.

## Topography, soil properties, and fine root data

Topographic data were taken from each plot of the different soil habitat types. GPS was used to record the longitude, latitude, and elevation of all plots. A clinometer was used to analyze the slope. The parameter settings followed *Harms et al. (2001)* and *Valencia et al. (2004)*.

The composition of soil organic matter (SOM) was measured by the potassium dichromate volumetric method (*Bao, 2000*). Water content was measured according to a soil agricultural chemistry analytical method (*Bao, 2000*). A soil auger were used to collect the O horizon soil (five samples in a diagonal line for each plot), and the O-elements (O horizon soil elements) were measured by inductively coupled plasma mass spectrometry (Agilent Technologies, Palo Alto, CA, USA). A soil thermograph (TZS-1W; Zhejiang Tuopu Instrument, Zhejiang, China) was used to record the temperature of each soil sample.

Fine root biomass was determined by rinsing the surface of the root samples with clean water. Then, the roots were dried in the shade. Vernier calipers were used to sift out root diameters >2 mm in all root samples. The fine roots of each sample were weighed fresh and biomass was determined by the formula (biomass = fine root fresh weight $\times\ 10^4/\pi^*$(inner diameter of earth drill/2)$^2$).

## Soil DNA extraction and miseq sequencing step

Soil total DNA was extracted from 50 mg of soil in each plot using the Fast DNA Isolation Kit according to the manufacturer's instructions (Qbiogene, Heidelberg, Germany). After checking DNA concentration and purity, the extracted soil DNA was stored at $-20\,°C$ for further use. The DNA extraction methods for the eukaryotic soil microbes were described by *Wu et al. (2011)* and *Bates et al. (2013)*.

The V4 region was defined by the primer set forward (5′-CCAGCASCYGCGGTAATTCC-3′) and reverse (5′-ACTTTCGTTCTTGATYRA-3′) to amplify the sequences (18S rRNA gene fragment). This primer set has been successfully used for eukaryotic microbes in several studies (*Guillou et al., 2013*; *Logares et al., 2014*). We used a two-step polymerase chain reaction (PCR) to construct the data library. First, specific primers were used to amplify the target fragment, and the target fragment was recycled with a Gel Recovery Kit (ASJ0013 Gel extraction AxyPrep DNA; Axygen Scientific Inc., Union City, CA, USA). Then, the recovery product was used as the template for the second PCR amplification. The purpose of this method was to add the necessary sequencing joints, barcodes, and sequencing primers to the ends of the segment for sequencing on the Illumina platform. More detailed steps of the two-step PCR are shown in Table S1 and S2. The libraries were sequenced using the PE300 and a MiSeq v3 Reagent Kit (Tiny Gene Co., Shanghai, China).

## Bioinformatics method

Mothur v. 1.33.3 and UPARSE (version v8.1.1756) software was used to analyze the sequence libraries following the methods of *Schloss et al. (2009)*. The original FASTQ files

were denoised and split by the barcode. The portable executable reads for each sample were run in Trimmomatic (version 0.35), using the parameters (maxambig = 0, length = 200–580, which protected the longer correct fragments through a higher threshold, maxhomop = 8) and avoided low quality base pairs. Moreover, the sequence data were aggregated with 97% homology to operational taxonomic units (OTUs) (*Edgar, 2013*), and the NCBI database was used to BLAST each OTU representative sequence. The OTUs were also determined based on NCBI (the highest score in BLAST was classified as the matched species, and incorrect OTU information was corrected or removed).

## Statistical procedures

Statistical analyses of alpha diversity (including OTU richness, Coverage, Chao 1, and Shannon's indices) were performed in Mothur v. 1.33.3 software following *Schloss et al. (2009)*. The topography, soil properties, and fine root data were analyzed by one-way analysis of variance using SPSS v. 19.0 software (SPSS Inc., Chicago, IL, USA). The eukaryotic community tree with the four soil habitat types was prepared using MEGA software (version 5.0) (Fig. S1). We used statistical software to analyze the dissimilarity of the dominant soil eukaryotic microbial groups among each soil habitat type, and a Venn diagram of shared OTUs was also prepared using the "VennDiagram" library; the indicator species analysis was prepared using the "indicspecies" library (Ri386 3.4.4). The network was generated with Cytoscape v. 2.8 software (*Smoot et al., 2011*). A principal component analysis was performed using Ri386. 3.4.4 (Table S3). Distance-based redundancy analysis (db-RDA) is an efficient method to detect soil eukaryotic microbial group-environmental correlations regarding the response of the soil eukaryotic community to environmental variables. Thus, the Monte Carlo permutation and db-RDA tests were charted using Ri386. 3.4.4 ("vegan" library), and the graph was optimized using Canoco software (Windows 4.5 package) (*Braak & Smilauer, 2002*). All raw sequences have been deposited into the NCBI database under accession number SRP148713.

# RESULTS

## Soil eukaryotic microbial data analyses

We obtained a total of 362,717 sequences and 1,056 OTUs (at 3% evolutionary distance) from the entire dataset, and the number of OTUs ranged from 396 to 603 per soil sample. The average coverage estimator value was 98.3% (range, 97.5–98.7%), indicating that the sequence data sufficiently covered the diversity of the eukaryotic population. Based on the Shannon's index analysis, the eukaryotic diversity in the study region ranged from 1.17 to 3.5 (Table 1). Tukey's HSD test was used to evaluate the significance of the results. Eukaryotic richness in the forest soil habitat type was significantly higher than the richness of the farmland, shrub, and grass soil habitat types. The diversity of eukaryotes in the forest and shrub soil habitat types was significantly higher than that in the other habitat types; farmland had the lowest eukaryotic diversity (Table 2). The indicator species analysis revealed soil eukaryotic microbe characteristic ($P < 0.05$) were different among the soil habitat types; a total of 98 indicators were detected in forest soil, rhizarian

**Table 1 Overview of soil eukaryotic microbes, sequence data, environmental factors, and other information.**

| Variables | Minimum | Average value | Maximum |
|---|---|---|---|
| Sequences | 19,941.00 | 30,226.00 ± 1291.00 | 35,045.00 |
| Number of OTUs | 396.00 | 491.00 ± 23.00 | 603.00 |
| Chao1 (richness) | 495.17 | 557.74 ± 17.67 | 651.43 |
| Soil eukaryote diversity | 1.17 | 2.37 ± 0.27 | 3.50 |
| Coverage (%) | 0.98 | 0.98 ± 0.002 | 0.99 |
| Water content (%) | 15.51 | 22.33 ± 1.78 | 32.90 |
| Soil organic matter (g kg$^{-1}$) | 13.50 | 19.91 ± 1.15 | 25.92 |
| Soil temperature (°C) | 12.23 | 13.49 ± 0.23 | 14.75 |
| Elevation (m) | 238.26 | 317.88 ± 12.17 | 387.88 |
| Slope (°) | 0.87 | 9.59 ± 2.68 | 31.89 |
| Fine roots biomass (g m$^{-2}$) | 12.13 | 28.56 ± 3.41 | 43.10 |

Note:
Diversity was measured by Shannon's index.

**Table 2 Geochemical characteristics of the soil samples and other sequence data information in the present study.**

| Soil type | Chao1 | Shannon's index | Coverage | Water content | SOM | Soil temperature | Elevation | Slope | Fine roots biomass |
|---|---|---|---|---|---|---|---|---|---|
| Forest | 643.3[a] | 3.21[a] | 0.975[b] | 16.3[c] | 19.9[ab] | 12.9[a] | 319[a] | 5.6[a] | 42.7[a] |
| Farmland | 500.7[c] | 1.17[c] | 0.988[a] | 32[a] | 14.9[b] | 13.3[a] | 284[a] | 4.2[a] | 12.9[d] |
| Shrub | 563.4[b] | 3.1[a] | 0.984[a] | 21.1[b] | 22.5[a] | 13.9[a] | 312[a] | 16.7[a] | 35[b] |
| Grass | 523.5[bc] | 2.0[b] | 0.987[a] | 19.9[b] | 22.4[b] | 13.8[a] | 356[a] | 11.9[a] | 23.6[c] |

Note:
Capital direction symbols (eg., a, b, c) indicate full (5%) significance.

taxa (35 indicators) had a dominant position; 49 indicators were detected in farmland soil, metazoa taxa (19 indicators) comprised the main group; 74 indicators were found in shrub soil (25 indicators in metazoa taxa), and 48 indicators were observed in grass soil (35 indicators in rhizarian taxa) (Table S4). More detailed information about the OTUs in all soil samples is shown in Table S5.

## Soil eukaryotic microbial composition south of the Taihang Mountains

The species composition of soil eukaryotic microbes remarkably differed among the different soil habitat types. Fungal (mainly Ascomycota, Basidiomycota, Chytridiomycota, Cryptomycota, and Glomeromycota), Cercozoa, Chlorophyta, Amoebozoa, Stramenopiles, Alveolata (mainly Ciliophora, Apicomplexa, and Dinophyceae), and Metazoa (mainly Nematoda) were the dominant soil eukaryotic microbial groups. OTUs that could not be identified were assigned as unclassified (Fig. 1). Within the soil fungi groups, Cryptomycota occupied the largest component, particularly in farmland (11.3–13.1%). Nematoda and Arthropoda dominated the Metazoa in the study area. Nematoda was extremely abundant among the soil metazoan groups (12.7–31.5%).

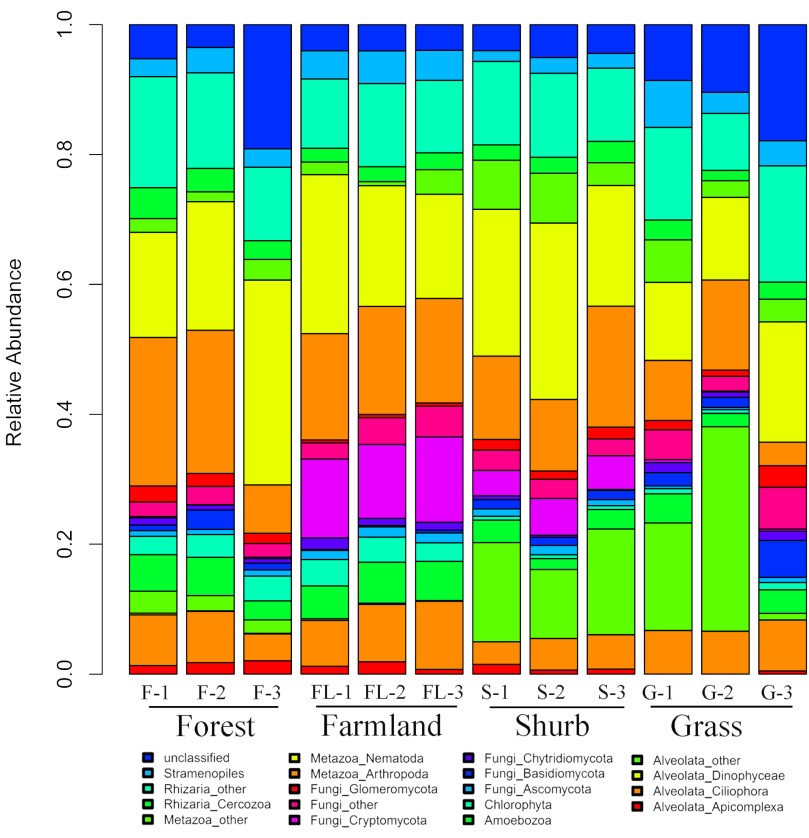

**Figure 1** **Relative abundances of the dominant soil eukaryotic microbial groups at each of the 12 individual sampling locations in the area south of the Taihang Mountains in a semi-arid region of China.** Percentage results were classified at the phylum level.

Eucoccidiorida was comprised mainly of order Apicomplexa sequences. Heterocapsaceae was the main group of the Dinophyceae sequences.

## Similarities and differences in the soil eukaryotic microbes among the different soil habitat types

We applied Venn and network diagrams to reveal the differences in the OTU distributions among the different soil habitat types. The Venn diagrams showed that the number of soil habitat type OTUs ranged from 552 (farmland soil) to 814 (grass soil), and only 270 of 1,056 OTUs were shared by the four soil habitat types. Moreover, there were 32 shared OTUs in the soil metazoan groups (200 OTUs), 23 shared OTUs in soil fungi (126 OTUs), 30 common OTUs in Alveolata (105 OTUs), and 90 OTUs existed in all of the rhizarian communities (268 OTUs) (Fig. 2). Network analyses revealed that the dominant OTUs were different among the four soil habitat types. Forest soil was dominated by OTU_7 (Metazoa), OTU_9 (unclassified), OTU_10 (Alveolata), and OTU_12 (Metazoa). Shrub soil was dominated by OTU_3 (Alveolata), OTU_17 (Metazoa), and OTU_22 (Fungi). Farmland soil was dominated by OTU_31 (Metazoa), whereas grass soil was dominated by OTU_20 (unclassified) (Fig. 3). Furthermore, network analyses provided more insight into these shared OTUs (OTUs existed in different land types). The results show that soil

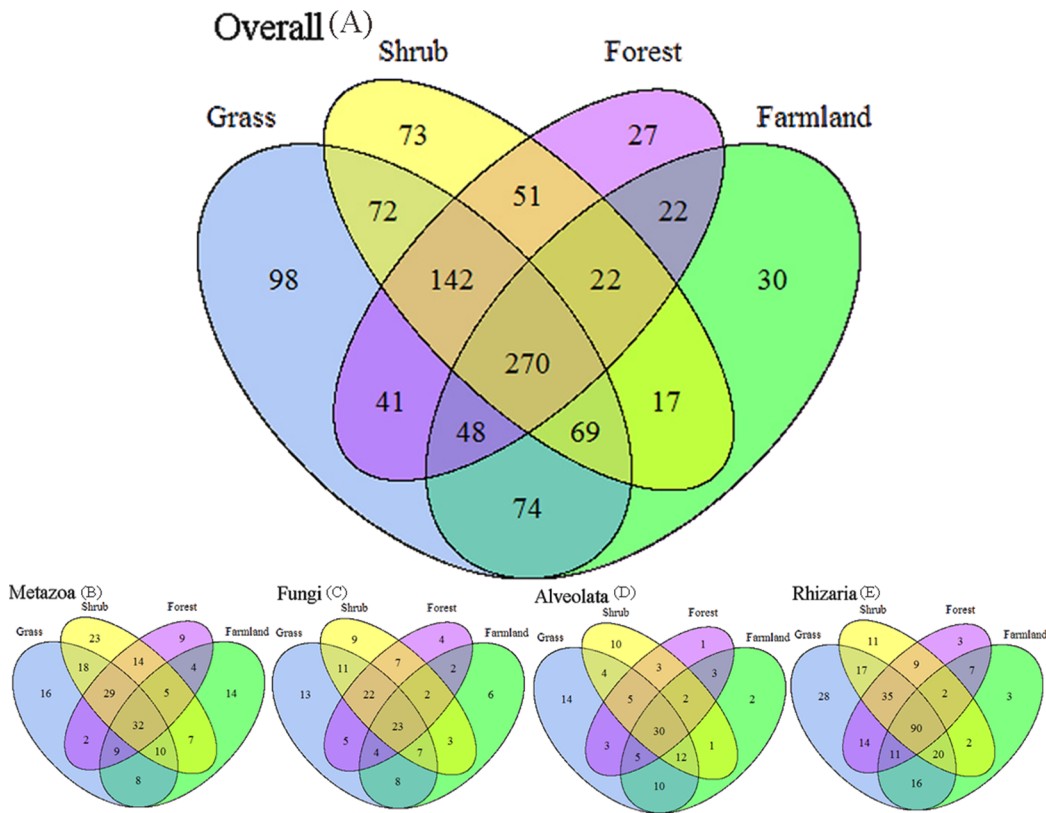

**Figure 2 Degree of overlap of the eukaryotic operational taxonomic units in the four soil habitat types.** (A) Overall. (B) Metazoa. (C) Fungi. (D) Alveolata. (E) Rhizaria.

habitat types with high species diversity (forest and shrub habitat types) shared larger OTUs (a higher number of sequences) than in the low species diversity habitat types (grass and farmland habitat types). The number of sequences of shared OTUs in the grass type was usually smaller than the others. The OTUs with few sequences (<200) was the main portion of shared OTUs among the four soil types, suggesting that the eukaryotic microbes with few sequences can more easily adapt to different soil habitats (Fig. 3).

## Effect of environmental factors on the soil eukaryotic community

The db-RDA of all soil habitat types and eukaryotic communities mainly showed the variation in the composition of the soil eukaryotic groups, which was explained by soil properties (water content, organic matter, O-elements, and soil temperature,) and partly by plant factors (fine root biomass) and geographical position (elevation and slope) (Fig. 4). These environmental factors explained 81.06% of the variation in the soil eukaryotic community, while 18.94% of the variation was not explained in the ordination diagram. TOM 1 ($r^2 = 0.7311$, $P < 0.01$), elevation ($r^2 = 0.6032$, $P < 0.01$), water content ($r^2 = 0.4706$, $P < 0.05$), and SOM ($r^2 = 0.5092$, $P < 0.05$) were the most important factors (Table 3), whereas the effects of soil temperature, slope, and fine root biomass were less influential.

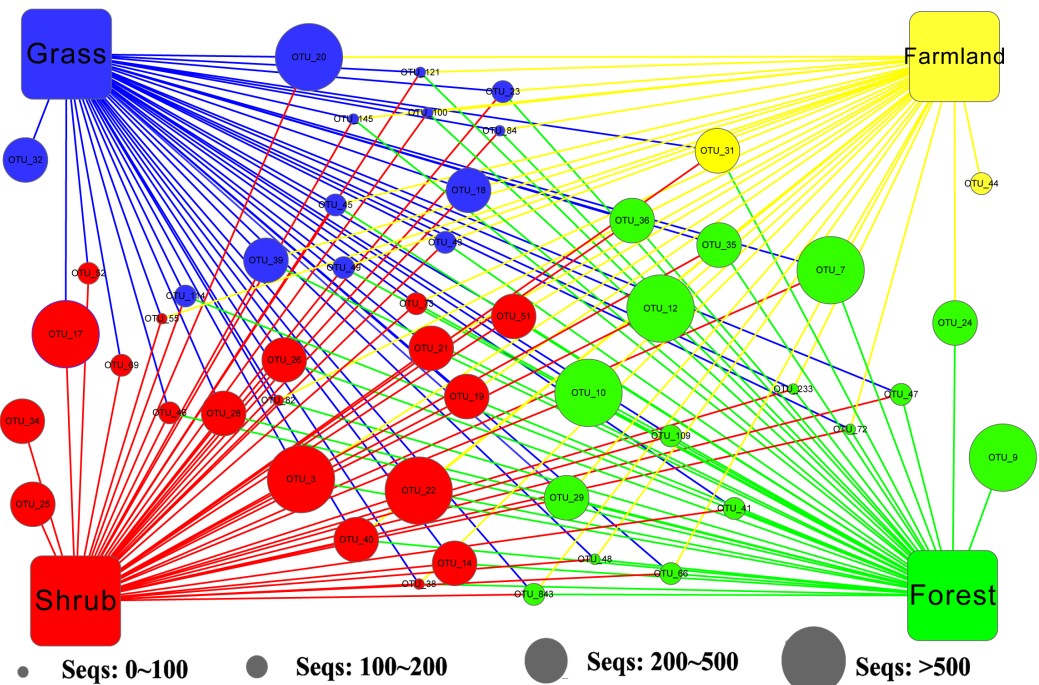

**Figure 3 Network of the dominant 50 operational taxonomic units (OTUs) in the different soil habitat types.** Square nodes of different colors represent the soil habitat types, whereas circular nodes represent the OTUs that connect to different soil habitat types through edges (lines). The center color of the circular node represents the specific soil habitat type that had the highest number of sequenced OTUs among the four soil habitat types.

# DISCUSSION

Previous studies have reported the soil eukaryotic taxa by molecular techniques at other sites (*Bailly et al., 2007*; *Tzeneva et al., 2009*). Our present study fully examined the soil eukaryotic microorganism data across different soil habitat types in the area south of the Taihang Mountains. The results show that the Shannon and chao1 indices were significantly different among the different soil habitats ($P < 0.05$); the highest value were in the forest soil habitat type, followed by shrub and farmland. This result may be because forest soil has more substrate diversity leading to species-rich microbes, and human disturbance might reduce the diffusion and growth processes of microbes (*Opik et al., 2010b*; *Jing et al., 2014*). Indeed, it is generally accepted that a single cropping system and excessive fertilization are applied to Chinese farmland ecosystems (*Xin et al., 2016*), so the microbial community structure tends to be less diverse in farmland. The forest and shrub soil habitat types provide sheltered habitat for small animals, and their activities and excreta increase soil nutrient contents to enrich the soil microbe species (*Corstanje et al., 2007*; *Raiesi & Beheshti, 2015*).

Sequencing results have elucidated that the dominant composition of eukaryotic groups among the different soil habitat types was roughly similar. Within the eukaryotic communities, the SAR (Stramenopiles, Alveolates, Rhizaria) supergroup (comprising about 39.76% of the sequences) was the dominant group at the phylum level, followed by

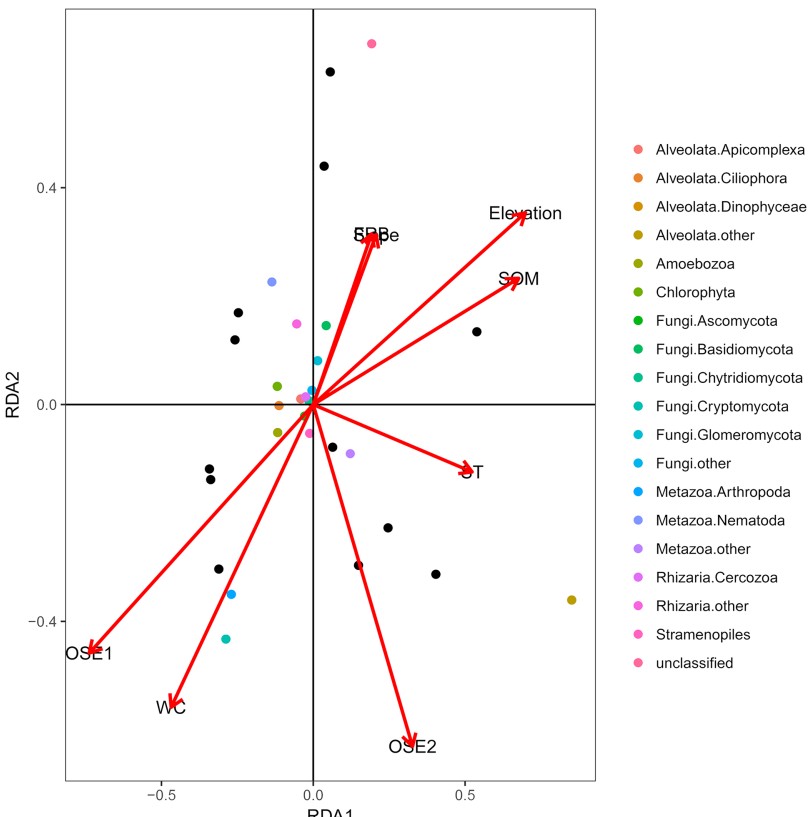

**Figure 4 Distance-based redundancy (db-RDA) tests used to interpret the correlations between the soil eukaryotic microbes and environmental properties.** Colored and black dots represent eukaryotic communities and plots, respectively. Arrows represent environmental factors. The length of an arrow indicates that the correlation is larger or smaller. The diagram was prepared using the most abundant eukaryotic communities. Abbreviations: WC, water content; SOM, soil organic matter; ST, soil temperature; OSE, O-elements; FRB, fine root biomass.

**Table 3 Monte Carlo permutation test to detect the relationship between the eukaryotic communities and the environmental properties.**

| Variables | RDA1 | RDA2 | $R^2$ | $P$-value |
|---|---|---|---|---|
| WC | −0.76896 | −0.63929 | 0.4706 | 0.046* |
| SOM | 0.95132 | 0.3082 | 0.5092 | 0.041* |
| ST | 0.99743 | −0.07172 | 0.2638 | 0.274 |
| Elevation | 0.91968 | 0.39266 | 0.6032 | 0.009** |
| Slope | 0.70586 | 0.70835 | 0.1198 | 0.582 |
| FRB | 0.6832 | 0.73023 | 0.1139 | 0.577 |
| O-elements 1 | −0.89505 | −0.44597 | 0.7311 | 0.002** |
| O-elements 2 | 0.54795 | −0.83651 | 0.3337 | 0.124 |

Notes:
The variable abbreviations are the same as in Fig. 4.
* $P < 0.05$.
** $P < 0.01$.

Metazoa (37.78%). Fungi was another highly represented group (12.82%). A similar distribution was reported in the Loess Plateau area of northwest China (*Jing et al., 2014*). *Bailly et al. (2007)* analyzed the eukaryotic microbes using a metatranscriptomic approach in a coastal sand dune. Those authors assumed that sequence number was closely related to cytoplasmic volume (biologically active), and most protists are single-celled organisms, compared with fungi, with a high copy number per haploid genome (*Bailly et al., 2007*). Therefore, using ribosomal sequences leads to a far higher ratio of protist to fungal sequences.

Furthermore, contrary to *Logares et al. (2014)*, who confirmed that several microbial taxa are unique to a single location and few are shared among different plots, our results showed that a large number of eukaryotic communities (25.6% of all OTUs) were shared by the different soil habitat types. These results may have occurred because soils have more complex chemical and physical features than water and the adaptive strategies of eukaryotes in soil differ from those in the aquatic environment (*Pommier et al., 2007*; *Fuhrman et al., 2008*). Our results also show that few OTUs existed in one soil habitat type, possibly because certain microbes exist in specific habitats and need a specific climate, nutrients, soil texture, or other factors (*Jing et al., 2014*; *Song et al., 2017*). Interestingly, the number of protozoan (Alveolates 29.1%; Rhizaria 33.6) OTUs was larger than the fungal OTUs (18.3%), suggesting that the protozoa were better adapted to the semi-arid mountain environment than fungi. The network results also indicate that the eukaryotic microbes with different sequences may have different environmental adaptation strategies. However, we studied the distribution of eukaryotic communities only from the habitat angle. Other possible factors, such as spreading ability and the competitive relationships of eukaryotes, were not considered here.

The db-RDA indicated that the variation in species composition of the soil eukaryotic microorganisms was mainly explained by soil properties and geographical factors. Our study assessed the relationship between variations in the soil eukaryotic community and multiple environmental factors in the area south of the Taihang Mountains. The results showed that the composition of soil eukaryotic communities was closely related to water content, SOM, the O-elements, and elevation and the O-elements was the most significantly related ($P < 0.05$). The large number of litters in the O horizon soil indicated greater nutrient cycling and the accumulation of a considerable pool of nutrient elements in soil for use by the microbes (*Smolander et al., 2005*; *Kanerva & Smolander, 2008*). The results also indicated that elevation was the second most influential factor ($P < 0.01$). The effect of elevation on biological distribution was integral to terrestrial ecosystems (*Rahbek, 2005*; *Malhi et al., 2010*). The climate change caused by changes in elevation alters the living space of microbial communities, resulting in the close association between microbes and elevation (*Smith, Halvorson & Boltan, 2002*; *Shen et al., 2016*).

Furthermore, the water content and SOM also significantly influenced the eukaryotic groups ($P < 0.05$) in this study. Water is an indispensable substance for life and can directly affect reproductive or metabolic processes, and indirectly influence the ecological niches of local microorganism or their physiological status (*Skopp, Jawson & Doran, 1990*;

*Liu et al., 2010*). In addition, *Christian et al. (2008)*, *Antisari et al. (2011)* and *Ludwig et al. (2015)* confirmed that soil SOM is an essential energy source for microbial activities including reproduction. In general, our research considered substrate variables as well as soil properties and environmental factors based on niche theory, but further studies on the relationship between geographic isolation or other factors and eukaryotic groups will be conducted in the future.

# CONCLUSIONS

Elucidating the drivers and assemblages of soil eukaryotic microbes is a critical step to reflecting the trends in recent ecological restoration. Species diversity and composition of soil eukaryotic microbes in the area south of the Taihang Mountains was delineated for the first time in this study. Our results indicate that the soil eukaryotic microbes were closely related to different soil habitat types. The highest eukaryotic richness and diversity were found in the forest soil habitat type; the SAR supergroup and Metazoa were the dominant soil eukaryotic groups, about 26% of total OTUs were shared in each soil habitat type. Our study also show the relationship between soil eukaryotic microbes and different environmental factors. The results suggest that the O-elements, water content, SOM, and elevation were significant driving factors in the soil eukaryotic microorganism communities. There findings confirm that forest soil habitat is an efficient means to restore local vegetation and shed new light on the distribution of local soil eukaryotic microbes in semi-arid areas.

# ACKNOWLEDGEMENTS

All authors thank the Xiaolangdi Ecological Station for the provision of the soil materials and testing ground. Also, we thank the Tiny Gene Bio-Tech (Shanghai) Co., Ltd. for their high-throughput sequence technology.

## Funding

This work was supported by the National Natural Science Foundation of China (31270750). The funders had no role in study design, data collection and analysis, decision to publish, or preparation of the manuscript.

## Grant Disclosures

The following grant information was disclosed by the authors:
National Natural Science Foundation of China: 31270750.

## Competing Interests

The authors declare that they have no competing interests.

## Author Contributions

- He Zhao conceived and designed the experiments, prepared figures and/or tables, authored or reviewed drafts of the paper, approved the final draft.
- Xuanzhen Li performed the experiments, authored or reviewed drafts of the paper.

- Zhiming Zhang analyzed the data.
- Yong Zhao conceived and designed the experiments.
- Peng Chen contributed reagents/materials/analysis tools, prepared figures and/or tables.
- Yiwei Zhu contributed reagents/materials/analysis tools.

## Data Availability

All raw sequences are available at the NCBI database under accession number SRP148713.

## Supplemental Information

Supplemental information for this article can be found online at http://dx.doi.org/10.7717/peerj.6042#supplemental-information.

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
