# Peer review of "Drivers and assemblies of soil eukaryotic microbes among different soil habitat types in a semi-arid mountain in China"

_PeerJ, doi:10.7717/peerj.6042_

## Round 0.1 · original submission · Major Revisions

Please consider carefully the all of the comments and suggestions from each reviewer as you prepare a revised submission. It is very important for the hypothesis, methods, results, discussion and conclusions to be clearly stated and supported by the data included in your manuscript. At the present the manuscript does not accomplish this.

·

Basic reporting

Basic Reporting
Introduction
Lines 42-45: this text is not clear
Lines 73-74: revise text, does not flow well from the text before
Line 80: revise the use of the connector However
Literature: about 40% of the articles cited are older than 2008. It is important to report newer and updated research reports
Materials and Methods
Lines 108-111: what is the purpose of this text? Is not connected with the rest of the paragraph
Line 124: regarding figure S1, are the authors of this article the creators of the image? Also, the legend description is poorly descriptive. Moreover, in the figure there are two soil depths shown and, in the methodology, only one is mentioned
Line 150: check xxtraction misspelling in legend for table S1
Line 155: check reference format
Results
Lines 184-185: revise the text regarding OTU, it is not clear and does not flow well
Line 193: regarding table 2, revise number of decimals reported. Be consistent on the number of decimals reported for each metric
Discussion
Line 263: check the use of the word however
Figures and Tables
Table 1: revise superscript in data units
Table 2: use a more descriptive table legend
Figure S1: improve quality of soil profile figure and state the authorship. Clarify the sampling depth shown in the figure and that expressed in methodology
Table S1: revise typo in the legend
Table S3: explain what is meant by indicator species analysis

Experimental design

Experimental design
Materials and Methods
Lines 93-95: if climatic conditions are provided, precipitation should be also included. This data is especially relevant given the climatic context of this work
Line 95: soil classification does not follow the nomenclature used by the USA Soil Taxonomy classification system, please revise and correct
Line 113: only one sampling depth is mentioned. It is not clear how this sampling strategy is related to the humus layer (see comment on the use of this terminology). This needs to be clarified, as it has implications on results such as that provided in Figure 4
Lines 123-124: explain this method better. How could you get water content from a dry sample?
Line 125: revise the use of the term humus layer. This terminology is not technically correct. Are you specifically referring to an O horizon or rather generalizing about a surface layer enriched in organic matter? I strongly suggest the correct use of edaphic terms
Lines 128-132: what was the area and soil depth for root sampling?
Line 156: what does PE stand for?

Validity of the findings

Validity of the findings
Results
Lines 193-195: text regarding indicator species analysis is not clear. This sentence is not informative
Lines 217-220: the text regarding network analysis does not add information to that already inferred from Venn diagrams
Discussion
Lines 238-340: this result is somehow expected, results should be more elaborated
Lines 247-250: same as comment above
Conclusions
Conclusion does not add novelty and it seems to be merely an abstract of the results presented through the text

Additional comments

The manuscript titled “Drivers and assemblies of soil eukaryotic microbes among different revegetation types in a semi-arid mountain in China” investigated the effect of different plant species used in soil restoration efforts at a site south of the Taihang Mountains in China, of semi-arid climate.
General comments
This work needs to be improved and a research hypothesis has to be included before publication. For further explanation, see specific comments below:
Abstract
Line 15: explain what would be meant by diversity and abundance
Line 19: why such comparison between protozoa and fungi? It is not clear the purpose of it
Introduction
Line 27: better specify to what is refer in the following sentence “Eukaryotic microorganisms are the dominant form of life” It this referred to biomass?
Line 30: revise the use of the word humus. In the context of the text it would be better to use the term organic matter. See article by Cotrufo et al 2015. Nature Geoscience for more insight into organic matter formation
Line 40: both articles sited took place in aquifer environments, this should be stated in the text to give an environmental context and contrast to the manuscript
Lines 40-42: in the text it is mentioned several studies have reported effects of organic fertilizers on eukaryotic communities. However, only one article has been cited.
Lines 44: what is sandtone 8?
Lack of hypothesis: this manuscript does not propose a hypothesis to test
Discussion
Lines 234-236: it is not clear what is the point of this sentence
Lines 237-238: this sentence is vague
Lines 258-262: it is not clear the connection of this statement with the discussion provided above in the same paragraph
Lines 269-272: what do the authors mean by “single OTUs”? are these singletons? If so, I would recommend removing these from the data set. Reasons on doing so are explained in the SOP by Schloss et al 2009
Lines 284-287: is this humus layer (see comment on terminology) present only in the forest soil? If so, this should be clearly explained starting from the methodology

Reviewer 2 ·

Basic reporting

No comment.

Experimental design

There are many errors in experimental design. For example, four dominant species Platycladus orientalis, Quercus variabilis, Robinia pseudoacacia and Lespedeza bicolor in forest, but the sampling design did not considered different species in the 12 independent plots, so the mixture sample can not represent the forest vegetation. On the other hand, changes in plant cover, increases in litter inputs, abundance of shrubs etc could explain diversity and composition of soil eukaryotic microbes, not only the SOM and fine root biomass.

Validity of the findings

No comment.

Additional comments

In this study the authors want to evaluate the species diversity and composition of soil eukaryotic microbes in artificial forest, artificial bush, farmland, and natural grassland at the south of the Taihang Mountain, China. My overall impression about this manuscript is negative. However, I also believe that there are plenty of aspects that can be improved for further study. Moreover, it needs careful round of editing, and careful going over the language by a professional, native speaker. There are too many awkwardnesses and grammatical errors.
Line 10-11. Rewrite for clarity. A variety of revegetation types include artificial forest and artificial bush, rather than farmland and natural grassland.
Line 36-45 and line 52-59. These description are not explicitly stated as a major goals for this study, this looks out of place.

---

## Round 0.2 · Major Revisions

Thank you for your resubmission. The manuscript is improved, however, as the review provided indicates we will still require major revision to the manuscript before publication. Please pay particular attention to the reviewer’s comments especially regarding the network analyses of the Illumina sequencing data.

In addition to responding to the reviewer’s comments, please consider the following:

It is not clear what is meant by “term organic matter” that is discussed in the abstract and throughout the Results section.

In Figure 3 the shapes are squares not rectangles and the size of the figure should be increased since the labels are not clear to a reader without magnification. This would help clarify some of the connections you are trying to establish between the soil habitats studies.

·

Basic reporting

The re- submitted manuscript titled “Drivers and assemblies of soil eukaryotic microbes among different re vegetation types in a semi-arid mountain in China” need further revision before publication.

Pay special attention to comments on methodology and results.

See comments on authors responses below

Experimental design

5.This sentence includes supplementary information for the sampling area. If you think it is not connected with the rest of the paragraph, we will delete the sentence. We have deleted the part of introduction.(Line 110-111)
we established study plots, which were the main component of the ecological systems classified as the forest, shrub, grass, and farmland soil habitat types in October 2017
Do you mean?
we established study plots, which represented were the main component of the ecological systems classified as the forest, shrub, grass, and farmland soil habitat types in October 2017

6.The purpose of this picture was to introduce the sampling tool. We apologize for the poor picture quality, and we have deleted the picture. The soil sample and root samples were collected at a soil depth of 5–10 cm. We have revised the sampling methods.(Line 113-117)
Soil samples were collected at a soil depth of 5–10 cm and were taken along the “S” model from five sampling points, then pooled together as one large sample for each plot (understood). After cleaning up the impurities, 200 g term organic matter in the shallow soil was also collected (this is still not clear, what is the depth for this sample? how do you collect organic matter from a soil?). Root samples in the vicinity of the soil samples were also collected with a soil auger (inner diameter = 4cm) at a depth of 5–10 cm.

20. Soil water content = [(fresh soil quality) − (quality after drying)]/( fresh soil quality) × 100%

The water measurement method has been revised. (Line 126-128)
Thoroughly dried soil was used to calculate soil water content by comparing the quality difference between dry and fresh soil (This is called gravimetric water content and the way it is was calculated is incorrect. Please see a recognized soil science book, for example Nature and Properties of soil to get the correct calculation method) . A soil auger were used to collect the term organic matter (five samples in a diagonal line for each plot), and the elemental analysis of term organic matter layer (is this referring to an organic horizon?) was measured by inductively coupled plasma mass spectrometry (Agilent Technologies, Palo Alto, CA, USA).

Validity of the findings

Findings need deeper argumentation

Additional comments

This manuscript is not yet ready for publication. There are some clarifications and corrections needed in methodology. Also, a deeper argumentation of network analyses would improve novelty of findings

---

## Round 0.3 · Minor Revisions

Please edit the manuscript one last time for punctuation errors and especially the missing hyphens throughout on word breaks between lines. The overall manuscript was much improved and with careful editing should be acceptable for publication.

---

## Round 0.4 · accepted · Accept

Thank you for resubmitting your revised manuscript. It is improved in scope, editorial corrections were made as suggested and it is acceptable for publication in PeerJ - the Journal of Life and Environmental Sciences. Congratulations!

#